# Mineralogical control on methylotrophic methanogenesis and implications for cryptic methane cycling in marine surface sediment

Ke-Qing Xiao [1✉], Oliver W. Moore [1], Peyman Babakhani[1], Lisa Curti [1] & Caroline L. Peacock [1]

Minerals are widely proposed to protect organic carbon from degradation and thus promote the persistence of organic carbon in soils and sediments, yet a direct link between mineral adsorption and retardation of microbial remineralisation is often presumed and a mechanistic understanding of the protective preservation hypothesis is lacking. We find that methylamines, the major substrates for cryptic methane production in marine surface sediment, are strongly adsorbed by marine sediment clays, and that this adsorption significantly reduces their concentrations in the dissolved pool (up to $40.2 \pm 0.2\%$). Moreover, the presence of clay minerals slows methane production and reduces final methane produced (up to $24.9 \pm 0.3\%$) by a typical methylotrophic methanogen—*Methanococcoides methylutens* TMA-10. Near edge X-ray absorption fine structure spectroscopy shows that reversible adsorption and occlusive protection of methylamines in clay interlayers are responsible for the slow-down and reduction in methane production. Here we show that mineral-OC interactions strongly control methylotrophic methanogenesis and potentially cryptic methane cycling in marine surface sediments.

[1] University of Leeds, School of Earth and Environment, Leeds LS2 9JT, UK. ✉email: k.q.xiao@leeds.ac.uk

The balance between the degradation and preservation of organic carbon (OC) in the oceanic environment is fundamentally important for the modulation and regulation of atmospheric $CO_2$ and $O_2$[1]. Despite decades of research, the factors that control this balance are still unclear[2,3]. One of the major hypotheses for slowing degradation and promoting preservation involves the protection of OC from microbial remineralisation via its adsorption to fine grained reactive minerals in marine sediments[4–7]. Although mineral protection is increasingly believed to protect OC from microbial remineralisation in soils and sediments for hundreds to thousands of years[4,7], a direct link between mineral adsorption and retardation of microbial remineralisation is often presumed[8]. Moreover, the link between mineral adsorption and inhibition of microbial remineralisation has rarely been investigated for a specific microbial remineralisation pathway, for which the adsorption and remineralisation of specific OC compounds by specific marine sediment microbes can be tracked and investigated at a microbially and mineralogically mechanistic level.

Methane ($CH_4$) production by methanogenic archaea is one of the main terminal steps in OC remineralisation in marine sediments[9], mediating 28.6% of global subseafloor OC degradation[10]. Recently it is shown that a cryptic methane cycle exists in the upper sediment layers[11,12], where unexpected methane production occurs in sulfate-rich marine surface sediment[11–15]. The turnover rate of this cryptic process in marine surface sediment is comparable to the methane benthic flux out through the sediment-water interface, and thus the cryptic methane cycle is significant in controlling methane exchange between sediment and seawater[11], which can affect the global greenhouse gas budget[9] and potentially oxygen dynamics throughout Earth history[16,17]. Methane production in this cryptic cycle is dominated by methylotrophic methanogenesis using small, methylated OC compounds like monomethylamine (MMA), dimethylamine (DMA) and trimethylamine (TMA). Intriguingly while these methylamines (MAs) are ubiquitous in marine sediments as important carbon and nitrogen sources for different microbes[9,18,19], they are present at only low concentrations in pore waters (sometimes below detection limits) but high concentrations in the solid phase, and show high affinity to marine sediment and clay minerals[20–24]. Early experiments show that MAs in particular can be adsorbed by clay minerals (montmorillonite and kaolinite)[22] and that the adsorption is affected by clay type, salinity and organic content[21]. These studies suggest that the concentration of MAs in marine sediment pore waters might therefore be moderated by adsorption of MAs onto clay minerals[20,21,23], but whether the presence of clays can control methylotrophic methanogenesis in marine surface sediment is unknown.

Here we explore the interactions between MAs, clay minerals and methanogenic archaea simultaneously to elucidate the link between adsorption and inhibition of remineralisation of OC, for an important microbial pathway in marine sediment. We hypothesise that the adsorption of MAs to clay minerals buffers the concentration of MAs in marine sediment pore waters and thus limits their accessibility to methanogenic archaea and so retards their microbial remineralisation. Three MAs (MMA, DMA and TMA) and the four most common clay minerals in marine sediment (montmorillonite, chlorite, illite and kaolinite)[25] are chosen for experiments, and *Methanococcoides methylutens* TMA-10 is used as a representative methylotrophic methanogen. *M. methylutens* is mesophilic and originally isolated from marine sediment, and methanogens belonging to this genus are ubiquitously detected and often dominant in marine surface sediment[11,26]. We perform adsorption experiments, microbial inoculation experiments, and use near edge X-ray absorption fine structure spectroscopy and X-ray diffraction to determine the mechanistic interactions between OC, minerals and microbes.

## Results

**Clay addition decreases free methylamines in solution.** The addition of clay minerals to solutions of MAs in a background matrix of growth medium without *M. methylutens* results in an increase of MAs adsorbed with the solid mineral phases and follows the trend chlorite < kaolinite < illite < montmorillonite (Fig. 1). The adsorption of the three MAs to all four clay minerals increases in the order MMA < DMA < TMA (Fig. 1). As a result of MA uptake by the clays there is a corresponding decrease in the concentrations of free MAs in the medium, where montmorillonite addition results in a decrease of free MAs by $9.2 \pm 0.08\%$ (MMA), $11.1 \pm 0.3\%$ (DMA) and $27.1 \pm 0.05\%$ (TMA) of the initial free MAs concentration. In the concentration range (0.03–30 mM) studied for each MA, isotherms are linear. Partition coefficients $K_{ads}$ (mL g$^{-1}$) for the MAs to the clays are therefore determined from the slope of the isotherms by linear fitting of the amount of MAs with the solid phase ($\mu$mol g$^{-1}$) to the amount still dissolved (mM) at equilibrium[22], as summarised in Supplementary Table 1. Results provide quantification of the affinity of the MAs to the clays in the order chlorite < kaolinite < illite < montmorillonite (Supplementary Table 1), and for the increasing affinity of the MAs in the order MMA < DMA < TMA (Supplementary Table 1). Results are in good agreement with those using $^{14}$C-MAs and a seawater matrix[22]. Based on the common occurrence of montmorillonite in marine sediments[25] and its ability to adsorb MAs, this clay is chosen to study methanogen-clay interactions.

**Clay addition inhibits methane production.** The addition of montmorillonite to solutions of MAs in growth medium containing *M. methylutens* results in a slow-down of methane production by *M. methylutens* for all three MAs compared to the montmorillonite free control, and this phenomenon becomes more noticeable with increased clay addition from 10 g L$^{-1}$ to 40 g L$^{-1}$ (Fig. 2). The final methane concentrations are also significantly ($P < 0.05$) reduced by montmorillonite addition, compared to the montmorillonite free control (Fig. 2). The slow-down of methane production and reduction in final methane production increase in the order MMA < DMA < TMA (Fig. 2). Specifically, the final methane production with addition of 10 g L$^{-1}$ of montmorillonite is reduced by $3.6 \pm 0.07\%$ for MMA, $4.0 \pm 0.2\%$ for DMA and $7.8 \pm 0.3\%$ for TMA, with addition of 20 g L$^{-1}$ montmorillonite is reduced by $8.0 \pm 0.05\%$ for MMA, $9.3 \pm 0.3\%$ for DMA and $13.9 \pm 0.1\%$ for TMA, and finally with addition of 40 g L$^{-1}$ montmorillonite is reduced by $14.9 \pm 0.2\%$ for MMA, $21.2 \pm 0.04\%$ for DMA and $24.9 \pm 0.3\%$ for TMA, compared to the montmorillonite free control.

**Clay addition partitions methylamines into different pools.** In a similar manner to the clay addition experiment in the absence of *M. methylutens* (Fig. 1) the addition of montmorillonite in the presence of *M. methylutens* decreases the concentrations of free MAs in the initial medium compared to the montmorillonite free control (Fig. 3). The decrease in free MAs increases with increased montmorillonite addition from 10 g L$^{-1}$ to 40 g L$^{-1}$, with a maximum decrease of $40.2 \pm 0.2$ % relative to the initial free MAs concentration (Fig. 3). In contrast to control experiments without *M. methylutens* (Supplementary Fig. 1), at the end of the incubation (after 144 h), there are no detectable free MAs in the medium (<0.5 $\mu$M) and no detectable exchangeable MAs in the solid for any of the treatments (Fig. 3), where exchangeable MAs are operationally defined as those that can be removed from

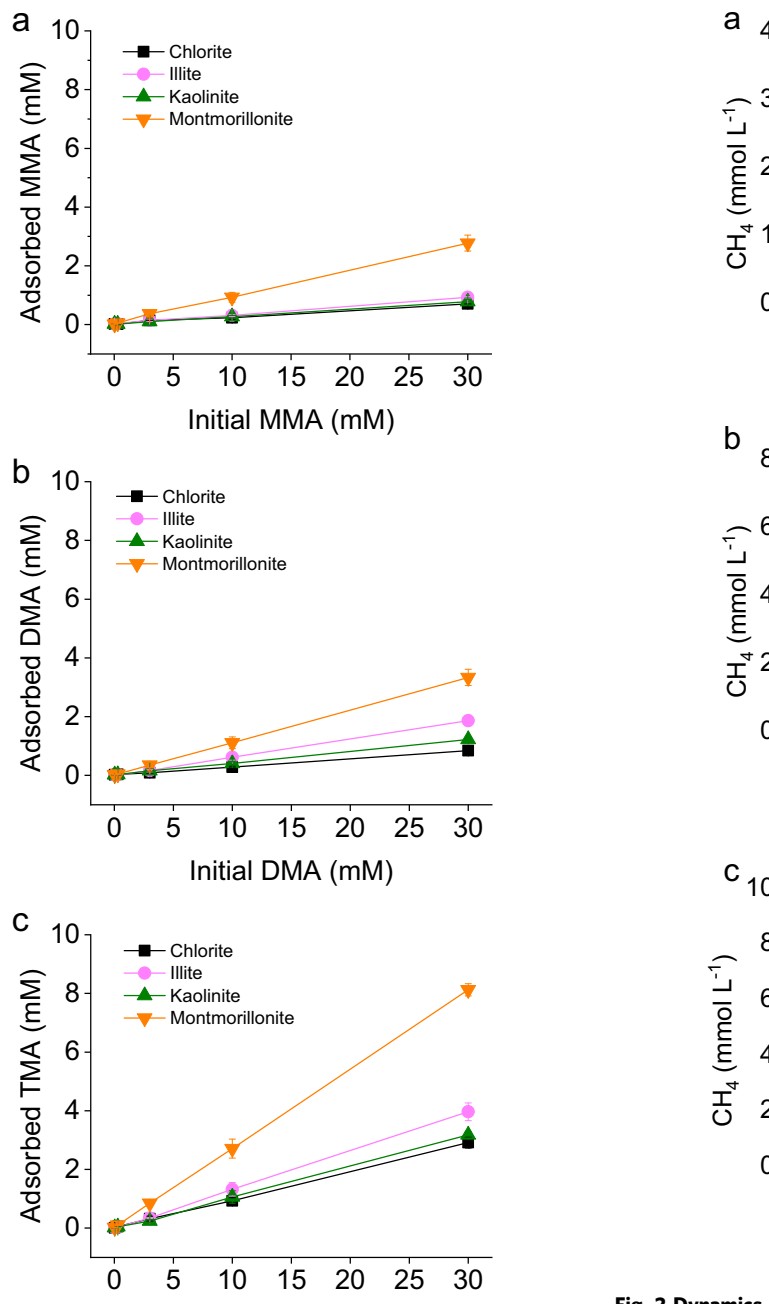

**Fig. 1 Adsorption of monomethylamine (MMA) (a), dimethylamine (DMA) (b) and trimethylamine (TMA) (c) to four clay minerals (20 g L⁻¹) in the absence of *Methanococcoides methylutens* TMA-10.** Initial concentrations of methylamines are 0.03, 0.3 (overlapping data points), 3, 10 and 30 mM; data points are presented as mean and error bars as standard deviation of triplicate samples.

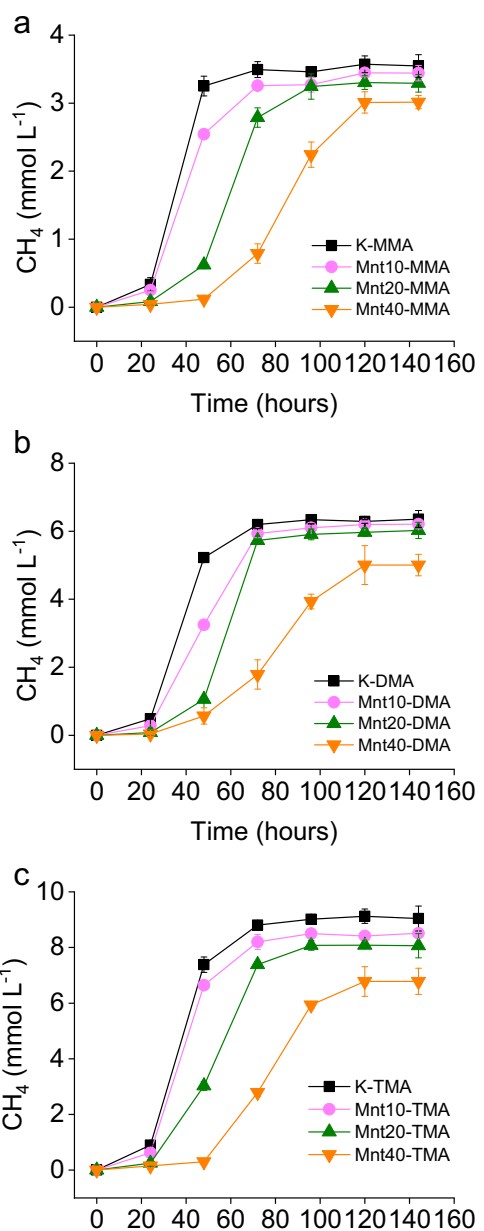

**Fig. 2 Dynamics of methane concentrations in the headspace of incubation bottles containing *Methanococcoides methylutens* TMA-10.** Monomethylamine (MMA) (**a**), dimethylamine (DMA) (**b**) and trimethylamine (TMA) (**c**); addition of montmorillonite at 0 (K), 10 (Mnt10), 20 (Mnt20) and 40 (Mnt40) g L⁻¹; data points are presented as mean and error bars as standard deviation of triplicate samples.

the solid phase using 1 M LiCl[21]. The fraction of MAs that are strongly adsorbed to the solid and are thus unavailable for exchange can be liberated during HF-HCl digestion which completely destroys the phyllosilicate structure[21]. After digestion the final strongly adsorbed MAs increase in the order MMA < DMA < TMA (Fig. 3). Specifically, the final non-exchangeable MAs with addition of 10 g L⁻¹ of montmorillonite are 2.6 ± 0.1% for MMA, 2.7 ± 0.2% for DMA and 3.6 ± 0.3% for TMA, with addition of 20 g L⁻¹ montmorillonite are 5.3 ± 0.04% for MMA, 5.4 ± 0.3% for DMA and 8.1 ± 0.01% for TMA, and finally with

addition of 40 g L⁻¹ montmorillonite are 6.8 ± 0.02 % for MMA, 10.0 ± 0.5% for DMA and 15.5 ± 0.1% for TMA, of the initial MAs in the montmorillonite free control (Fig. 3).

**Chemical interaction of methylamines with montmorillonite.** The addition of montmorillonite to solutions of MAs without *M. methylutens* results in a strong chemical interaction between MAs and the mineral surface (Fig. 4). For all three MAs adsorbed by montmorillonite, analysis by near edge X-ray absorption fine structure spectroscopy shows the amino N peak region at ~405.5–409.7 eV[27] is reduced in amplitude and shifted to lower energy compared to their respective free MA standard. The

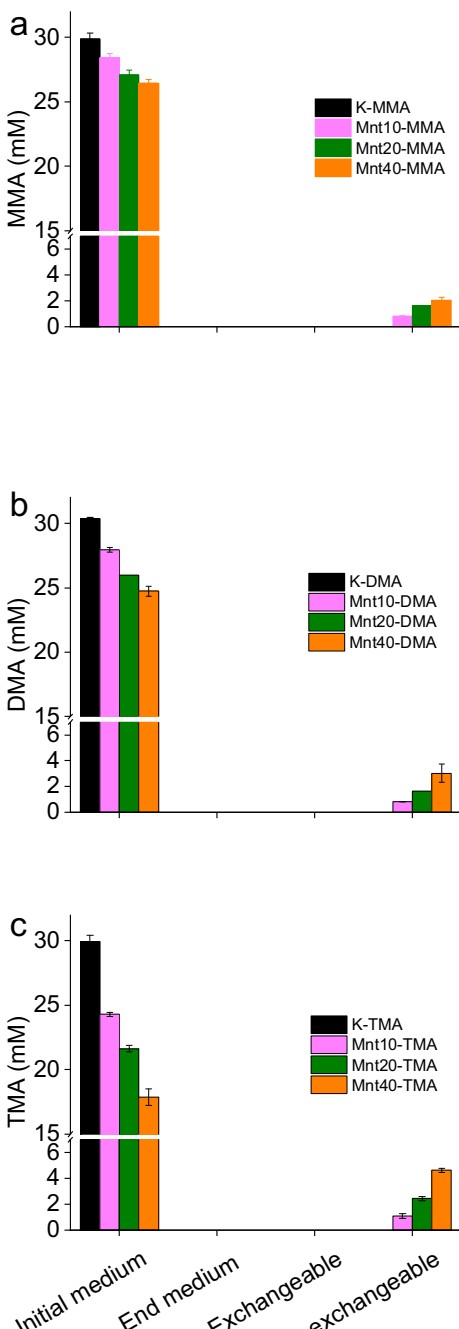

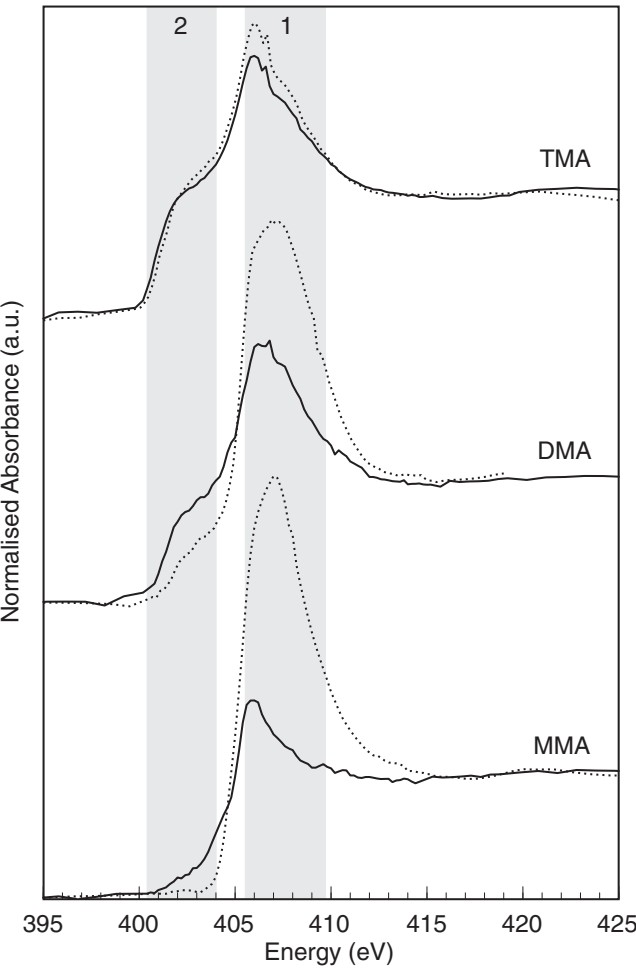

**Fig. 4 N 1s NEXAFS spectra for methylamines adsorbed onto montmorillonite in the absence of *Methanococcoides methylutens* TMA-10.** Data plotted as energy (eV) vs. normalised absorbance (arbitrary units). Grey bands show the spectral regions associated with amino N (band 1) and the shoulder feature present on the low energy side of the amino N peak (band 2). For each methylamine the free methylamine standard spectrum (dotted lines) and the montmorillonite associated sample spectrum (solid lines) are presented. MMA stands for monomethylamine, DMA for dimethylamine and TMA for trimethylamine. Spectra are stacked with an arbitrary vertical offset for clarity.

**Fig. 3 Distribution of Monomethylamine (MMA) (a), dimethylamine (DMA) (b) and trimethylamine (TMA) (c) in different pools in the presence of *Methanococcoides methylutens* TMA-10.** Free methylamines in initial medium after 24 h equilibration before inoculation of strain TMA-10 (initial medium), free methylamines in medium after 144 h (end medium, below detection limit < 0.5 μM), exchangeable methylamines adsorbed by montmorillonite after 144 h extracted using 1 M LiCl (exchangeable pool, below detection limit < 0.5 μM) and non-exchangeable methylamines adsorbed by montmorillonite after 144 h extracted using 5 M HF-1 M HCl (non-exchangeable pool, values are transformed into concentrations related to initial medium volume 10 mL for comparison). MMA stands for monomethylamine, DMA for dimethylamine and TMA for trimethylamine; addition of montmorillonite at 0 (K), 10 (Mnt10), 20 (Mnt20) and 40 (Mnt40) g L$^{-1}$; data are presented as mean and error bars as standard deviation of triplicate samples.

reduction in peak amplitude and energy shift is greatest for MMA-montmorillonite and follows the order MMA-montmorillonite > DMA-montmorillonite > TMA-montmorillonite. For all three MAs adsorbed by montmorillonite there is also a distinct low energy feature at ~400.5–404 eV that manifests as a shoulder on the low energy side of the main amino N peak. This shoulder is absent in the free MMA standard and becomes progressively more developed in the order MMA < DMA < TMA in the free MA standards and MA-montmorillonite samples, relative to the amino N peak amplitude.

**Interaction between *M. methylutens* and montmorillonite.** The addition of montmorillonite to solutions of MAs in growth medium containing *M. methylutens* results in a mineralogical change in montmorillonite after incubation, where, in agreement with previous work[28,29], X-ray diffraction patterns show that there are no new mineral phases formed but the 001 peak of montmorillonite shifts to lower *d*-spacing and becomes wider and less intense (Supplementary Fig. 2). Concomitantly there is a

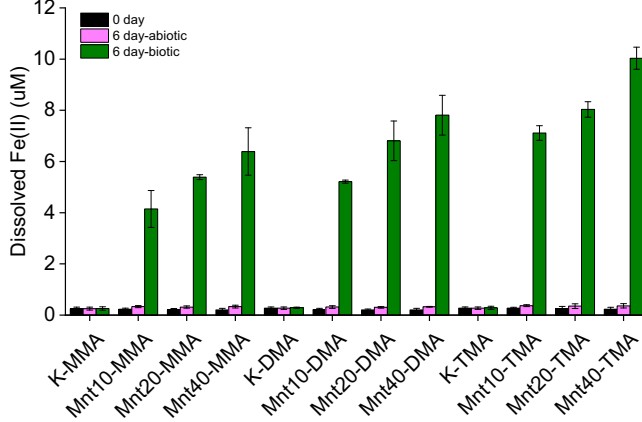

**Fig. 5 Concentrations of dissolved Fe(II) in solution in the presence (biotic) and absence (abiotic) of *Methanococcoides methylutens* TMA-10 at 0 and 144 h.** MMA stands for monomethylamine, DMA for dimethylamine and TMA for trimethylamine; addition of montmorillonite at 0 (K), 10 (Mnt10), 20 (Mnt20) and 40 (Mnt40) g L$^{-1}$; data are presented as mean and error bars as standard deviation of triplicate samples.

significant ($P < 0.05$) increase in dissolved Fe(II) concentrations in the medium, compared to the montmorillonite free control and the control experiments without *M. methylutens* (Fig. 5). Due to the adsorption of Fe(II) to clay and the existence of sulfide in the medium, it is not possible to quantitatively estimate total Fe(II) released in the reaction system.

## Discussion

Whilst the protective nature of minerals for OC is a well-established hypothesis, a direct link between adsorption and remineralisation is often presumed, and has rarely been tested for specific adsorption mechanisms or specific microbial remineralisation pathways[8]. Here we investigate the link between adsorption and remineralisation of MAs during the recently discovered cryptic methane cycle in marine surface sediments[11,12] in which methylotrophic methanogens utilise MAs as non-competitive substrates for methane production[12,13]. In these surface sediments MAs are present at only low concentrations in pore waters but high concentrations in the solid phase, leading previous work to suggest that MAs pore water concentrations are moderated by adsorption of MAs onto clays[20–24]. Here we hypothesise that the adsorption of MAs by clays can limit the availability of MAs to methylotrophic methanogens and thus inhibit their remineralisation, such that adsorption provides a strong control on methylotrophic methanogenesis and potentially the cryptic methane cycle in marine surface sediments.

In our study, we find that clay mineral addition (at concentrations 10–40 g L$^{-1}$) results in the adsorption of MAs to the solid phase and thus significantly decreases the concentrations of free MAs in the medium (Fig. 1). We also find that clay mineral addition results in a slow-down of methane production and a reduction in the final absolute concentration of methane produced by the methylotrophic methanogen *M. methylutens* (Fig. 2). To determine the interactions between MAs, clay and *M. methylutens* and how these processes control methane production, we operationally define the different solid-phase pools of adsorbed MA (Fig. 3) and investigate the MA-clay system (Fig. 4) and the MA-clay-*M. methylutens* system (Fig. 5) at a mechanistic level.

Previous work indicates that OC adsorbed to clays are protected from desorption by extracellular enzymes[30,31], and thus the supply of MAs to methanogens in the presence of clays should be limited

by their equilibrium desorption from the clay particles. In our work, a reversible adsorption of MAs to montmorillonite that limits their supply to *M. methylutens* (Fig. 1) can explain the slow-down of methane production in the presence of clay (Fig. 2). Reversible adsorption then results in an exchangeable pool of MAs (up to $24.7 \pm 0.3\%$) that are utilised by *M. methylutens* (Fig. 3) and an absence of free MAs (below detection limit) in the medium (Fig. 3). We also find however, a significant non-exchangeable pool of MAs (up to $15.5 \pm 0.1\%$) at the end of the experiment (Fig. 3). This relatively non-reversible adsorption of MAs must also limit their supply to *M. methylutens* (Fig. 1) but in addition, can explain the majority (45.6–74.6%) of the reduction in the final amount of methane produced (Fig. 2 and Supplementary Fig. 3, which shows that there is a linear relationship between the amount of non-exchangeable MAs and the reduction in the final methane production). Our results show that adsorption of MAs to clay exerts a first-order control on the methylotrophic methanogenesis of MAs at clay concentrations and experimental conditions of ionic strength and pH approximating marine surface sediments. A slow-down of methane production in the presence of clay up to ~2 days (Fig. 2) is comparable to the turnover rates of methane in marine surface sediments[11]. Our results therefore suggest that the adsorption of MAs to clay could substantially affect the cryptic methane cycle and thus local methane cycling in marine surface sediments. The fact that no detectable MAs are present in the medium and a relatively large fraction is adsorbed to clay (Fig. 3), also offers a mechanistic explanation for the very low concentrations of MAs in marine surface sediment pore waters (in μmol L$^{-1}$ or often below detection limits) but high concentrations in the solid phase[13,20–24].

Previous work to determine the precise adsorption mechanisms between MAs and clay is limited to wet chemical desorption experiments that cannot determine which specific chemical and / or physical processes dominate and control MAs uptake[22]. The dissociation constants (*p*Ka values) of MMA, DMA and TMA are 10.6, 10.7 and 9.8, respectively, which means they are predominantly positively charged (MMA: $CH_3NH_3^+$; DMA: $(CH_3)_2NH_2^+$; TMA: $(CH_3)_3NH^+$) in our medium and marine sediment pore waters[32,33]. It is therefore speculated that MAs adsorb to negatively charged clays in sediments[34] via reversible electrostatic bonds[22]. During the adsorption of amino acids to montmorillonite a peak amplitude reduction and shift to lower energy of the $\delta_s(NH_3^+)$ band is observed using Fourier transform infra-red spectroscopy, and is suggested to reflect reversible electrostatic and / or hydrogen bonds between positive amino groups and negative silanol adsorption sites[35]. In our work, either reversible electrostatic and / or hydrogen bonds can explain our similar amino N peak changes and the manifestation of the low energy shoulder (Fig. 4). This low energy shoulder is attributable to Rydberg character[27]. The shoulder is absent in the free MMA standard, where Rydberg mixing between the -NH bonds attenuates the Rydberg signal, but becomes progressively evident for the free DMA and TMA standards as the number of -NH bonds decreases[27]. As the -NH bond interacts with clay, there are fewer -NH bonds for valence mixing and the Rydberg peak is enhanced between each MA and their respective standard. The Rydberg peak is also progressively developed from MMA-clay, with the most -NH bonds, to TMA-clay, with the least -NH bonds. The decreasing extent of the peak changes from MMA to TMA however, indicates that the particular bonds associated with this phenomenon are weakest for TMA (Fig. 4). Electrostatic bonds weaken from MMA to TMA because progressive replacement of an H atom on the amino group by an electron-donating methyl group increases the negative charge on the amino group[36]. Hydrogen bonds and van der Waals forces between -CH groups and clay however, strengthen from MMA to

TMA, as the number of methyl groups (molecular weight) increases[37,38]. Our results therefore provide direct evidence of the reversible electrostatic bonding of MAs to clay, but indicate that hydrogen bonding is also required to explain increasing adsorption affinity from MMA to TMA (Fig. 1).

Whilst the slow-down in methane production (Fig. 2) is due to reversible adsorption via electrostatic and hydrogen bonds, this adsorption mechanism cannot entirely explain the differences in the adsorption affinity of our clays (Fig. 1). This is because if reversible adsorption was the only adsorption mechanism for MAs, these differences in adsorption affinity should be explained by the fact that montmorillonite has a higher surface area and thus a higher number of reversible electrostatic adsorption sites, compared to the other clays[39]. The partition coefficients $K_{ads}$ (mL g$^{-1}$) normalised to clay surface area $K_{area}$ (mL m$^{-2}$) however, show that montmorillonite still possesses elevated adsorption affinity (Supplementary Table 1). Furthermore, reversible adsorption via electrostatic and hydrogen bonds, cannot explain the reduction in final methane production (Fig. 2), which is partly due to a relatively non-reversible adsorption that increases from MMA to TMA (Fig. 3). In addition to reversible adsorption via electrostatic and hydrogen bonds, it is therefore clear that an additional adsorption mechanism, that is dependent on other mineralogical properties, plays an important role in MAs adsorption. Of the clays we investigate, montmorillonite is the only one with an interlayer of cations that are easily exchangeable (montmorillonite (Swy-2) cation exchange capacity 85 meq/ 100 g[40]). The small molecular size of MAs therefore makes it possible for some MAs to migrate into the interlayer of montmorillonite[41,42]. This increases the adsorption affinity of montmorillonite compared to the other clays, and creates a pool of MAs that are likely to be less readily exchanged than those located at the particle surfaces[22,38]. The fact that hydrogen bonds and van der Waals forces between -CH groups and clay strengthen from MMA to TMA likely favours their sequentially increasing adsorption into this non-exchangeable pool (Fig. 3). Our results therefore indicate that MAs are also occlusively protected from microbial remineralisation as a non-exchangeable pool located in clay interlayers.

Our results show that the reduction in final methane production (Fig. 2) is only partly due to relatively non-reversible adsorption via occlusion into clay interlayers (Supplementary Fig. 3, which shows that for every unit of non-exchangeable MAs there are >1 unts of reduced final methane production). Instead previous work shows that the bioreduction of structural Fe(III) that is commonly found in clay minerals (montmorillonite (Swy-2) contains 4.23% Fe$_2$O$_3$[43]) results in a release of Fe(II) into solution[28,29,44,45]. This subsequently inhibits methanogenesis for many methanogens, including *Methanosarcina barkeri* MS, *Methanosphaera cuniculi* 1R7, *Methanobacterium palustre* F, *Methanococcus voltaei* A3 and *Methanolobus vulcani* PL-12/M[46], *Methanospirillum hungatei*, *Methanosarcina barkeri*, *Methanosaeta concilii* and *Methanosarcina barkeri*[44] and *Methanothermobacter thermautotrophicus*[28]. For montmorillonite, bioreduction of structural Fe(III) results in the release of Fe(II) to solution[28,29,44,45] and a concomitant shift to lower $d$-spacing together with a broadening and reduction in intensity of the 001 peak in the X-ray diffraction pattern[28]. In our study bioreduction of structural Fe(III) by *M. methylutens* can explain the increase of dissolved Fe(II) in the medium (Fig. 5) and similar peak changes for montmorillonite (Supplementary Fig. 2) observed after incubation. Previous work indicates that the peak changes reflect a partial collapse of the montmorillonite interlayers, in which the proportion of interlayers with ~17 Å $d$(001) spacing decreases and those with ~13 Å $d$(001) spacing increases[28]. A partial collapse of the interlayers is unlikely to substantially affect the

reversible adsorption of MAs via electrostatic and hydrogen bonds or restrict the occlusion of molecules with small molecular size like MAs into the interlayers. The presence of Fe(II) however, is suggested to contribute to slowed methane production and inhibition of final methane production (Fig. 2) because either, the presence of montmorillonite increases the redox potential of the system, which is unfavourable for methanogens[28,46,47], and/or the diversion of electron flow to Fe(III) reduction inhibits methanogenesis[28,44,46]. Although diverse microbes are adversely affected by Al$^{3+}$, dissolved Al$^{3+}$ was not detectable in the medium (<0.1 μM) and thus the toxicity from Al$^{3+}$ is negligible in this study.

In conclusion, our study provides direct evidence that the adsorption of OC (MAs) by minerals (clay) limits the availability of OC (MAs) to microbes (*M. methylutens*) and results in a slow-down and reduction of remineralisation of OC (methane production). We provide a mechanistic understanding of the interactions between methylated substrates, minerals and methylotrophic archaea. Our results indicate that adsorption of MAs to clays provides a hitherto unrecognised first-order control on methylotrophic methanogenesis and potentially cryptic methane cycling in marine surface sediments. Our data show that methylotrophic methanogenesis is increasingly inhibited with increasing montmorillonite concentrations (from 10 to 40 g L$^{-1}$) (Fig. 2). Montmorillonite concentrations in marine surface sediments are reported to reach near 130 g L$^{-1}$ off the Coast of Peru in the South–East Pacific Ocean[21,48] and up to around 200 g L$^{-1}$ in pelagic sediments of the South Pacific Ocean[48]. As such, we suggest that methanogenesis might be further inhibited with increasing montmorillonite concentrations, and thus that this process might be most important in geographical locations with high montmorillonite, like the South Pacific Ocean. Given that the turnover rate of cryptic methane cycling in marine surface sediment is comparable to the methane benthic flux out through the sediment-water interface[11], the inhibition of this process potentially affects methane exchange between sediment and seawater and thus overall carbon cycle budgets. Further research is required however, to investigate the relationship between montmorillonite concentrations, inhibition of methylotrophic methanogenesis and carbon cycling, under more complex experimental conditions and in situ in marine surface sediments. We suggest that similar studies to ours here could also directly investigate mineral-OC interactions for different OC substrates, minerals and microbes involved in related biogeochemical processes.

## Methods

**Experimental design.** This study was conducted to investigate the link between mineral adsorption and microbial remineralisation during interactions between MAs, methanogens and clay minerals in marine sediment. Three methylamines (monomethylamine (MMA), dimethylamine (DMA) and trimethylamine (TMA)) and four common clay minerals in marine sediment (chlorite, illite, kaolinite and montmorillonite) were chosen for experiments, and *Methanococcoides methylutens* TMA-10 was used as a representative methylotrophic methanogen. We chose $n = 3$ for all experiments.

**Culturing of methanogenic archaea.** *Methanococcoides methylutens* TMA-10 (DSM 2657) was purchased from Deutsche Sammlung von Mikroorganismen und Zellkulturen (DSMZ) GmbH, and is an anaerobe, mesophilic archaeon that was initially isolated from marine sediment[49,50]. *M. methylutens* is an obligatory methylotrophic marine methanogen, the type species of its genus, while TMA-10 is the type strain. Recommended protocol from DSMZ was followed and standard anaerobic techniques[51] were done for culturing this strain. In brief, the growth medium consisted of (per liter) 0.34 g KCl, 4.00 g MgCl$_2$ x 6H$_2$O, 3.45 g MgSO$_4$ x 7H$_2$O, 0.25 g NH$_4$Cl, 0.14 g CaCl$_2$ x 2H$_2$O, 0.14 g K$_2$HPO$_4$, 18.00 g NaCl, 1.00 g Na-acetate, 2.00 g yeast extract, 2.00 g trypticase peptone, 5.00 g NaHCO$_3$, 3.00 g trimethylamine x HCl (31.3 mM TMA in final growth medium), 10.00 mL trace element solution, 2.00 mL Fe(NH$_4$)$_2$(SO$_4$)$_2$ x 6H$_2$O solution (0.1% *w/v*) and 10.00 mL vitamin solution. The growth medium was made anoxic by flushing with

$N_2/CO_2$ (80:20) gas mix and adding 0.50 g L-cysteine-HCl x $H_2O$, 0.50 g $Na_2S$ x 9 $H_2O$, with 0.50 mL Na-resazurin solution (0.1% $w/v$) as a redox indicator. The final pH of the growth medium was adjusted to 7.0–7.2, which was dispensed into 75 mL serum bottles and sterilised by autoclaving. *M methylutens* was then added to the growth medium and incubated at 30 °C.

The chemical composition of the culture medium mimics the chemical composition of seawater. The calculated ionic strength of the culture medium is ~0.5 M, approximating natural seawater (0.6–0.7 M), while pH is ~7.0, which is within the range of marine sediment pore water pH (6.9–8.3)[32,33]. The culture medium is optimised for cell growth under laboratory conditions[49,50] and is therefore nutrient-richer than pore water and contains a higher concentration of organic substrates (mM range) than pore water (typically μM range).

Control experiments were prepared to check for the inherent presence of MAs in the yeast extract (and/or trypticase peptone). In brief these control experiments involved only the sterile growth medium and the addition of *M. methylutens* in the absence and presence of TMA, and methane accumulation and cell growth were monitored over 144 h. No methane accumulation or cell growth was detected over the incubation period (near zero intercepts for both methane accumulation (~0.0001) and the growth curve (~0.01) (Supplementary Fig. 4)). As *M. methylutens* is an obligatory methylotrophic marine methanogen, it can only use small, methylated molecules, like MAs, for methane production[49], thus the absence of methane accumulation and cell growth in the experiments without TMA demonstrate that there were no MAs or other methylated compounds in the yeast extract.

**Abiotic adsorption experiments.** Four minerals from the Clay Minerals Society were used in the experiments: chlorite (#CCa-2), illite (#IMt-2), kaolinite (#KGa-2) and montmorillonite (#SWy-3). All clays were ground to a fine powder (flour-like) using a TEMA laboratory agate disc mill and sieved (45 μm) before use. The <45 μm size fraction was used for the experiments. The total carbon contents for the four clays were 0.031% for chlorite, 0.086% for illite, 0.011% for kaolinite and 0.17% for montmorillonite as determined by LECO SC-144DR (LECO Corporation, USA). The specific surface areas were 9.1 $m^2 g^{-1}$ for chlorite, 19.6 $m^2 g^{-1}$ for illite, 20.3 $m^2 g^{-1}$ for kaolinite and 29.6 $m^2 g^{-1}$ for montmorillonite according to a BET $N_2$ adsorption test[39] using a Gemini® VII 2390 Surface Area Analyzer (Micromeritics Instrument Corporation, USA). For the adsorption experiments, the sterile growth medium prepared without TMA was used as a matrix, and the four clays were sterilised by autoclaving. The four clays were then added separately into the growth medium reaching a final concentration of 20 gdw $L^{-1}$, which is within the average concentration ranges of these clays in marine surface sediments[48]. After pre-equilibration of the growth medium and clays, stock solutions of MMA, DMA or TMA (Merck KGaA, Germany) were added separately, reaching initial concentrations of 0.03, 0.3, 3, 10 and 30 mM. Adsorption isotherms of MAs adsorption to clays were determined at room temperature (25 °C). All vials were capped and shaken for 24 h[22]. After centrifugation (2760 × g, 5 min), the supernatant was taken from each vial to measure concentrations of MAs in solution using a ThermoScientific ICS5000 Ion Chromatography System. The amounts of MAs adsorbed by the clays were determined as the difference between MAs concentrations in solution initially added and after 24 h.

**Biotic methanogen-clay interaction experiments.** For the biotic methanogen-clay interaction experiments, the sterile growth medium prepared without TMA was used as a matrix, and montmorillonite was sterilised by autoclaving. For each experiment 10 mL of the growth medium was added into 75 mL serum bottles. Stock solutions of MMA, DMA or TMA (Merck KGaA, Germany) were added separately into the growth medium to reach 30 mM concentration and montmorillonite was added to make a slurry with final montmorillonite concentrations of 0 (K), 10 (Mnt10), 20 (Mnt20) and 40 (Mnt40) gdw $L^{-1}$. After pre-equilibration of the slurry for 24 h, 0.2 mL of inoculum of *M. methylutens* was transferred into the serum bottles. All solutions and the inoculum were transferred using sterile needles and syringes. Serum bottles were incubated at 30 °C in an incubator. All experiments were performed in triplicate. At given time points, 1 mL of headspace gas was collected from the serum bottles and analysed for $CH_4$ concentration. Before inoculation and at the termination of the experiments after 144 h, free MAs, dissolved Fe(II) and $Al^{3+}$ in the growth medium were measured in the supernatant after centrifugation of 0.5 mL suspension inside an anaerobic glove box (Coy Laboratory Products, Grass Lake, MI, USA). Separation of exchangeable and non-exchangeable MAs in montmorillonite was conducted sequentially[21]. Briefly after centrifugation exchangeable MAs in the remaining montmorillonite were extracted with 1 M LiCl for 24 h with continuous agitation, and the supernatant after centrifugation was analysed with Ion Chromatography. After extraction with LiCl, non-exchangeable MAs in montmorillonite were then extracted with 5 M HF-1 M HCl for 24 h, followed by evaporation to dryness. Boric acid was then added and evaporated to dryness overnight, and finally the sample was re-dissolved with deionised water and analysed with Ion Chromatography.

Abiotic (inoculum-free) control experiments were prepared both in the absence and presence of montmorillonite. In brief the clay-free control experiments involved only the sterile growth medium and the addition of MAs at 30 mM, and MAs concentrations in the medium were monitored at 24 h (initial medium) and

144 h (end medium). No reduction in MAs concentrations in the medium at 24 or 144 h was detected, thus demonstrating that loss of MAs via adsorption to the vials or degradation was negligible (Supplementary Fig. 1, experiment K shown with black bars). The clay-added control experiments involved only the sterile growth medium, the addition of MAs at 30 mM and the addition of montmorillonite at 10 (Mnt10), 20 (Mnt20) and 40 (Mnt40) gdw $L^{-1}$, and MAs concentrations in the medium were monitored at 24 h (initial medium) and 144 h (end medium), and in the exchangeable and non-exchangeable pools at 144 h. No significant change in MAs concentrations in the medium between 24 and 144 h was detected, thus demonstrating that 24 h was sufficient for equilibration (Supplementary Fig. 1, experiment Mnt10, Mnt20 and Mnt40 shown with pink, green and orange bars, respectively).

Use of 10–40 gdw montmorillonite $L^{-1}$ is based on the abundance and range of this clay in average marine surface sediments[48], and on measurements of the marine surface sediment mineralogy at the specific study sites where low pore water but high sediment MAs concentrations are observed[20–24]. Average marine surface sediments have a global range between ~2–195 gdw montmorillonite $L^{-1}$ in surface continental shelf sediments and ~65–215 gdw montmorillonite $L^{-1}$ in surface pelagic sediments[48], while the specific study sites range between ~8–129 gdw montmorillonite $L^{-1}$ [20–24,48].

**Analytical methods.** At the beginning and end of the experiments, pH values were measured by an Orion 8102BNUWP ROSS Ultra pH Electrode (Thermo Fisher Scientific, USA). The pH values in the incubation bottles remained nearly constant at around 7.0 throughout the experimental duration.

Methylamines were analysed using a ThermoScientific ICS5000 Ion Chromatography System (Thermo Fisher Scientific, USA) fitted with column CS16 4 μm (2 × 250) mm CG16 4 μm (2 × 50) mm (40 °C). The injection volume was 10 μL, the sampler tray was at 4 °C, and the gradient programme was: 15 mM methanesulphonic acid (MSA) for 30 min, increased at 5.33 mM MSA $min^{-1}$ to 70 mM MSA. Methylamine standards were prepared using the same MAs chemicals as used for the experiments, in the same matrixes, and were injected every 10 samples. Detection limits for MAs were ~0.5 μM and the RSD was <3%. Mass balance for MAs was calculated based on the MAs concentrations measured in solution at 24 h (initial medium) and 144 h (end medium), the MAs concentrations measured in solution after extraction with 1 M LiCl (exchangeable) and the MAs concentrations measured in montmorillonite after extraction with 5 M HF-1 M HCl (non-exchangeable), which were transformed into solution concentrations related to the initial medium volume 10 mL for comparison. For example, percentages of non-exchangeable MAs = non-exchangeable MAs/(MAs in initial medium + MAs in end medium + MAs in exchangeable pool + non-exchangeable MAs) * 100%.

Methane concentration was analysed on an Agilent 7890a Gas Chromatography System (Agilent Technologies, Inc, USA) connected to a flame ionization detector (FID) using HayeSep Q 80/100 column. The gas mixture for calibration was supplied by Air Liquide (France) and was injected every 20 samples. The RSD was <2%. Mass balance for methanogenesis from MAs was calculated following the reactions below:

$$4CH_3NH_3^+ + 3H_2O \rightarrow 3CH_4 + HCO_3^- + 4NH_4^+ + H^+ \qquad (1)$$

$$4(CH_3)_2NH_2^+ + 6H_2O \rightarrow 6CH_4 + 2HCO_3^- + 4NH_4^+ + 2H^+ \qquad (2)$$

$$4(CH_3)_3NH^+ + 9H_2O \rightarrow 9CH_4 + 3HCO_3^- + 4NH_4^+ + 3H^+ \qquad (3)$$

where for 1 mol of carbon in each MA, 0.75 mol carbon was transformed to $CH_4$, such that for 30 mM initial concentration of MMA, DMA and TMA, total $CH_4$ production in the clay-free biotic methanogen-clay interaction experiments was 0.75 * 30 mM, 2(0.75 * 30 mM) and 3(0.75 * 30 mM), respectively. The percentage of methanogenesis reduction was then calculated as: percentage of methanogenesis reduction = (methane production in clay-free treatment − methane production in each treatment) / methane production in clay-free treatment * 100%.

Dissolved Fe(II) in the medium was measured by Ferrozine assay and $(NH_4)_2Fe(SO_4)_2 \cdot 6H_2O$ solution was used as a standard every 20 samples, with a RSD of <2%[52,53]. Also dissolved $Al^{3+}$ was measured using the Inductively Coupled Plasma Mass Spectrometry (Thermo Fisher Scientific, USA). Standards were injected every 10 samples, and rhodium (m/z 103) at a concentration of 1 μg $L^{-1}$ was added to all standards and samples for use as an internal standard. The RSD was <2%. Precision was checked by triplicate injection of standards for each analytical method.

To investigate the mechanistic interactions between MAs and montmorillonite, N 1 s scanning transmission X-ray microscopy near edge X-ray absorption fine structure spectroscopy (STXM NEXAFS) spectra of the abiotic adsorption samples were recorded on Beamline I08, Diamond Light Source Ltd, Oxfordshire, UK. Approximately 2 mg of each sample were resuspended in ~500 μL of ultrapure water and sonicated. An aliquot of ~5 μL of suspension was then pipetted onto a silicon membrane window (Silson Ltd) and left to air dry. The membrane windows were glow discharged prior to loading with sample to improve particle distribution. Windows were then inserted into a high vacuum environment (<1 × 10⁻⁵ mBar) and analysed in scanning transmission mode. Stacked data sets for N were collected between 380–440 eV, at a energy resolution of 380–400 eV (1 eV), 400–415 eV

(0.2 eV), 415–420 (0.5 eV) and 420–440 eV (1 eV). To minimise beam damage on the sample, dwell times were set to 10 ms per energy step following beam damage tests conducted by repeatedly measuring the same area of sacrificial samples. Beam damage manifests as a N NEXAFS peak at an absolute energy of 400.4 eV[54]. Sacrificial spectra with beam damage were discarded but the position of the beam damage peak was used for absolute energy calibration by shifting all spectra in energy space by the required energy to align the beam damage peak to 400.4 eV. The dark signal was measured routinely by scanning a small area with the sample shutter closed. X-ray absorption stacks were aligned using the Axis2000 software. Spectra were extracted from the edge regions of individual coprecipitate particles and the dark signal was subtracted from the raw data using the Mantis software. Spectra were then exported for baseline correction, alignment, calibration and normalisation using the Athena software[55]. Baseline correction and normalisation avoid spectral dependence on the total N content, and as such, spectral features and peak shifts are indicative of N molecular structure and chemistry and not N concentration effects occurring during NEXAFS measurement. Peak identification for the normalised spectra was achieved with reference to literature assignments[27].

To evaluate mineralogical changes upon interaction of the methanogen with montmorillonite, X-ray diffraction was used to scan montmorillonite in the methanogen-clay interaction experiment before and after 144 h incubation with strain TMA-10. Smear mounts were prepared and scanned with a Bruker D8 X-ray Diffractometer (Bruker Corporation, USA) using CuKα wavelength, a Lynxeye detector, and a power of 1600 W (40 kV, 40 mA). The analytical conditions were as follows: a step size of 0.0197° per step, a counting time of 1 s per step, and a scanning range of 2–86° $2\theta$.

**Statistical analyses**. All data are expressed as the mean ± standard deviation ($n = 3$). Differences between means were evaluated by one-way analysis of variance. Significance was defined at the 0.05 level. All analyses were performed using SPSS 18.0.

**Reporting summary**. Further information on research design is available in the Nature Research Reporting Summary linked to this article.

## Data availability
All data generated in this study have been deposited to the general data repository Figshare and can be accessed at https://doi.org/10.6084/m9.figshare.17074676.

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

## Acknowledgements

This research project has received funding from the European Research Council (ERC) under the European Union's Horizon 2020 research and innovation programme (Grant agreement No. 725613 MinOrg, C.L.P.). Diamond Light Source provided access to beamline I08 (STFC grant number SP20839, C.L.P.) that contributed to the results presented here. We acknowledge Royal Society Wolfson Research Merit Award (WRM/FT/170005, C.L.P.) and the Environmental Mineralogy Group Early Career Researcher Bursary from the Mineralogical Society of Great Britain and Ireland (K.-Q.X.). We thank Andrew Hobson, Stephen Reid, Fiona Keay, Lesley Neve and David Ashley for laboratory support at University of Leeds. We also thank Clare Woulds and Mingyu Zhao for helpful comments and suggestions on the manuscript. K.-Q.X. also thank Bo Barker Jørgensen and Nils Risgaard-Petersen who have helped inspire this work.

## Author contributions

The original hypothesis was formulated by C.L.P. and K.-Q.X. K.-Q.X., P.B. and C.L.P. designed the experiments and K.-Q.X. performed the experiments. C.L.P., O.W.M. and L.C. collected the synchrotron data and contributed to its processing, and C.L.P., K.-Q.X., O.W.M. and L.C. contributed to its interpretation. K.-Q.X. and C.L.P. interpreted the data and wrote the manuscript, with contributions from all other authors.

## Competing interests

The authors declare no competing interests.
