## [Peer Review File · Nature Communications]

Mineralogical control on methylotrophic methanogenesis and implications for cryptic methane cycling in marine surface sedimentREVIEWER COMMENTS

Reviewer #1 (Remarks to the Author):

The manuscript “Mineralogical control on cryptic methane cycling in marine surface sediments” by Xiao et al. investigated the mechanism by which a mineral phase (montmorillonite clay) might hinder microbial respiration of sedimentary organic matter. This work is high quality and is an important part of the puzzle to understand sedimentary organic matter preservation. It will be of interest to a broad audience of oceanographers and geochemists and I believe that this manuscript may be appropriate for publication in Nature Communications after several issues are addressed. My major comments/questions are included below.

It would be useful for the authors to explain in the paper why montmorillonite is more effective at adsorbing MAs as compared to other clays (i.e. differences in surface area) when the results of different clays are presented.

The authors should be clear about how much montmorillonite is in the average marine sediments and how much its content ranges between different marine settings. This would provide important context for the choice of montmorillonite additions. Why was 40 g/L chosen? In some of the presented results (for example, Fig. 2 DMA) the difference is not very large at lower concentrations, so this choice should have a good justification.

I followed up both citations- one was a study of three marine sites off the coast of Washington (Keil et al., 1994). The numbers here are wt. % from a total sediment sample that was cored with a box corer, which is very different than a mass of minerals per volume of water. The other reference (Th. Leipe et al., 2000) does not appear to agree with the chosen values. In this paper, they had a “particulate matter” concentration of 4000 mg/L or 10x less than the chosen value. Furthermore, in this paper, smectite does not appear to be a dominant mineral phase.

Additionally, could the information about natural montmorillonite content and variation be used to upscale the results to indicate the scale of MAs that may be sorbed in a given area and how much potential methane production this prevents? This would significantly help readers evaluate the real-world impact of these experimental findings.

Can you clarify in the text how the culture medium compares to what might be expected for a natural marine system (ionic strength, pH, etc.)? In particular, how might competition for adsorption sites differ in the real world in the presence of more organic compounds, especially within sediment porewaters? This seems relevant particularly when trying to assess whether the inter-layer migration that is proposed (i.e. Line 252) will be significant within a natural setting.

What are the implications in a natural system of a slowdown of methane production on a scale of a day or two? Do the authors believe this would impact the local methane cycling? Based on the plots, the change in timing is more significant than the change in final production due to the addition of clay.

Reviewer #2 (Remarks to the Author):

Review of Xiao et al. “Mineralogical control on cryptic methane cycling in marine surface sediments” (NCOMMS-21-31580-T)

Synopsis

The primary focus of this study is to probe and quantify the mechanisms by which mineral protection may regulate organic matter (OM) preservation. In particular, the authors focus on methylotrophic methanogenesis and the role of mineral adsorption in regulating the concentration of methylamines (MAs), the dissolved substrate. Using a suite of batch incubation experiments (and abiotic controls), the authors show that MA adsorption to minerals both slows down methane production (by MA

adsorption-desorption reactions) and decreases the final methane concentration (by the formation of a non-exchangeable, clay-bound MA fraction). The authors conclude that this provides the first mechanistic insight into how mineral interactions protect OM, and they speculate how this may relate to other mineral-OM processes and the global carbon cycle.

Overall, I find this study to be highly topical and relevant to important carbon-cycle questions. (I note, however, that I am not a synchrotron expert and I therefore hope that other reviewers can better comment on this aspect of the manuscript.) It is well-written and articulated, and it poses a testable, mechanistic hypothesis for mineral-OM interactions. Still, I do have some “larger” comments and concerns, which I articulate below followed by line-item comments. Once these issues are fixed/clarified, I support publication of this manuscript in Nature Communications.

Please do not hesitate to contact me for further details regarding this review.

Sincerely, Jordon Hemingway jhemingway@ethz.ch

Reviewer #3 (Remarks to the Author):

Xiao and coauthors investigated the effect of mineral adsorption on the production of methane from methylamines (MAs), the process of which was reported earlier to drive the cryptic methane cycling in marine surface sediment. They found that MAs were strongly adsorbed into clay minerals. This adsorption significantly reduces methylamines concentrations in the dissolved pool and hence impacts methane production from a typical methylotrophic methanogen. They examined the adsorption of three MAs into four clay minerals and further tested the methane production by *Methanococcoides methylutens* TMA-10 with different concentrations of montmorillonite. This wonderful study provides the systematic evaluation of mineralogical control on microbial production of methane from MAs and improves our understanding in the interactions between physical, chemical and biological processes. However, the following comments need to be addressed before recommendation of publication.

General comments:

1. Title. The title is kind of misleading because the authors did not conduct experiments with any marine sediments. Although the methanogen *Methanococcoides methylutens* TMA-10 used in this study was initially isolated from marine sediment, there is no direct linkage between the results obtained from pure culture and cryptic methane cycling in marine surface sediments. I understand the authors intended to stress the potential implication, but this is not major finding that can be directly supported from the experiments. Also, methylotrophic methanogenesis can occur in both surface sediment and deeper sediments, and thus adsorption can affect methane production in deep sediment as well.
2. The above comment also arises another question. I wonder why the authors did not use marine sediments to test the adsorption of MAs and methanogenesis. I think the slurry they made was far away different from real sediments, e.g., the porosity, MAs concentrations.

Other comments:

Line 83: The MAs concentrations showed in Fig.1 were not the same as 0.03-30 mM here (also as described in Line 325).

Line 175: Unit is missing for dissolved Fe in Fig. 5.

Line 192: I am confused here. What is the range of particulate matter concentrations? Does particulate matter mean particulate carbon? I think the slurry made with 10-40 g L⁻¹ clay were much more diluted than marine sediments. Why the concentrations were comparable?

Line 270-273: This is speculative. I think no evidence existed for concurrent production and oxidation of methane by *Methanococcoides methylutens* TMA-10.

Response to Reviews

Reviewer #1 (Remarks to the Author):

The manuscript “Mineralogical control on cryptic methane cycling in marine surface sediments” by Xiao et al. investigated the mechanism by which a mineral phase (montmorillonite clay) might hinder microbial respiration of sedimentary organic matter. This work is high quality and is an important part of the puzzle to understand sedimentary organic matter preservation. It will be of interest to a broad audience of oceanographers and geochemists and I believe that this manuscript may be appropriate for publication in Nature Communications after several issues are addressed. My major comments/questions are included below.

1. It would be useful for the authors to explain in the paper why montmorillonite is more effective at adsorbing MAs as compared to other clays (i.e. differences in surface area) when the results of different clays are presented.

Response: Done. We thank the reviewer for this suggestion and have included an explanation of why montmorillonite is more effective at adsorbing MAs as compared to other clays, but we have included this in our discussion section (rather than when the results of the different clays are presented) because, following the suggestion of reviewer 2, we have normalised our K_{ads} partition coefficients to surface area in Supplementary Table 1) and it is clear that differences in surface area, and thus differences in the number of surface sites available for reversible adsorption of MAs, cannot entirely explain the differences in the adsorption affinity of our clays (Fig. 1). After normalisation to surface area montmorillonite still shows enhanced adsorption affinity compared to the other clays (Supplementary Table 1). This helps make our case that there is another adsorption mechanism occurring in addition to reversible adsorption of MAs – i.e., the relatively non-reversible adsorption that we measure (Fig. 3) and that we attribute to the occlusion of MAs in the clay inter layers. This adsorption mechanism is only able to occur with montmorillonite because it is the only clay that we investigate that has an interlayer of cations that are easily exchangeable. So the combined existence of reversible adsorption plus this non-reversible adsorption results in montmorillonite being more effective at adsorbing MAs. This extended discussion is now included at lines 260–282.

2. The authors should be clear about how much montmorillonite is in the average marine sediments and how much its content ranges between different marine settings. This would provide important context for the choice of montmorillonite additions. Why was 40 g/L chosen? In some of the presented results (for example, Fig. 2 DMA) the difference is not very large at lower concentrations, so this

choice should have a good justification. I followed up both citations- one was a study of three marine sites off the coast of Washington (Keil et al., 1994). The numbers here are wt. % from a total sediment sample that was cored with a box corer, which is very different than a mass of minerals per volume of water. The other reference (Th. Leipe et al., 2000) does not appear to agree with the chosen values. In this paper, they had a “particulate matter” concentration of 4000 mg/L or 10x less than the chosen value. Furthermore, in this paper, smectite does not appear to be a dominant mineral phase. Additionally, could the information about natural montmorillonite content and variation be used to upscale the results to indicate the scale of MAs that may be sorbed in a given area and how much potential methane production this prevents? This would significantly help readers evaluate the real-world impact of these experimental findings.

Response: Done. We thank the reviewer for asking for clarity on our choice of montmorillonite addition and apologise for our references at original line 193, which refer to particulate matter concentrations and are therefore difficult to relate to a mass of montmorillonite per volume of water. The major phyllosilicate clay minerals in marine sediments are smectite, chlorite, mica and kaolinite (Chamley, 1989), and hence we conducted our initial clay addition experiments using a smectite (montmorillonite), chlorite, mica (illite) and kaolinite (reference to Chamley, 1989 is now added at line 63). Following the results of these experiments indicating that MAs associate most readily with montmorillonite and based on the ubiquity of montmorillonite in marine sediments (Chamley, 1989), we then conducted our methane production experiments using montmorillonite. The methane production experiments were conducted at 40 gdw (grams dry weight)/L based on the abundance and range of this clay in average marine surface sediments (Griffin et al., 1968), and on measurements of the marine surface sediment mineralogy at the real-world study sites (cited in the main manuscript) where low porewater but high sediment MAs concentrations have been observed (see below):

Abundance and range of montmorillonite in average marine surface sediments:

So for example, in continental shelf sediments montmorillonite abundances range from **~2 gdw montmorillonite/L** in some North West Atlantic Ocean sites (sediments with ~2% clay of which on average ~16% is montmorillonite; then using an average porosity of marine surface sediments (~0.8 (Martin et al., 2015)) there are ~250 cm³ of sediment per 1 L of porewater, and using an average density of marine surface sediments (~2.7 gdw/cm³ (Tenzer and Gladkikh, 2014)) there are ~675 gdw of sediment per 1 L of porewater; thus if ~2% of this sediment is clay (~14 gdw clay/L) and 16% of this clay is montmorillonite, then there are ~2 gdw montmorillonite/L) to **~195 gdw montmorillonite /L** in the East Indian Ocean (sediments with ~70% clay of which on average ~41% is montmorillonite), while pelagic sediments of the Atlantic, Pacific and Indian Ocean range from **~65**

– **215 gdw montmorillonite/L** (sediments with ~60% clay of which on average between ~16 – 53% is montmorillonite) (Griffin et al., 1968).

Measurements of the marine surface sediment mineralogy at the real-world study sites:

- Lee and Olsen, 1984 – Buzzards Bay, North West Atlantic **~39 gdw montmorillonite/L** (sediments with similar clay content to those at Long Island Sound ~36% (Wang and Lee, 1990) of which on average 16% is montmorillonite (Griffin et al., 1968, Deep-Sea Research)); Eastern Tropical North Pacific (400 km off Mexico) **~45 gdw montmorillonite/L** (sediments with similar clay content to those in the Gulf of California ~19% of which on average ~35% is montmorillonite (Griffin et al., 1968)).
- Wang and Lee, 1990 – Long Island Sound, North West Atlantic **~39 gdw montmorillonite/L** (sediments with ~36% clay (Wang and Lee, 1990) of which on average 16% is montmorillonite (Griffin et al., 1968)); Coast of Peru, South East Pacific **~129 gdw montmorillonite/L** (sediments with ~36% clay (Wang and Lee, 1990) of which on average 53% is montmorillonite (Griffin et al., 1968)).
- Wang and Lee, 1990; Wang and Lee, 1993; Wang and Lee, 1994 – also studied the salt marsh sediments of Flax Pond, New York **~8 gdw montmorillonite/L** (sediments with ~7% clay (Wang and Lee, 1990) of which on average 16% is montmorillonite (Griffin et al., 1968)).
- Zhuang et al., 2017 – Aarhus Bay, Bay of Aarhus, Denmark **~54 gdw montmorillonite/L** (sediments with ~50% clay (Jensen and Bennike, 2009) of which on average 16% is montmorillonite (Griffin et al., 1968)).

Thus with a global range between ~2 – 195 gdw montmorillonite/L in surface continental shelf sediments and ~65 – 215 gdw montmorillonite/L in surface pelagic sediments, and a range between ~8 – 129 gdw montmorillonite/L at the sites specifically studied for MAs biogeochemistry, we feel that our range of 10 – 40 gdw montmorillonite/L for our methane production experiments is reasonable. Given that the truly marine sites specifically studied for MAs biogeochemistry (i.e., excluding salt marsh Flax Pond) range between ~39 – 54 gdw montmorillonite/L, we also feel that focusing our discussion on our 40 gdw montmorillonite /L results is justified. It should also be noted that our choice of 20 gdw/L montmorillonite, chlorite, illite and kaolinite in our initial clay addition experiments is also within the average concentration ranges of these clays in marine surface sediments (Griffin et al., 1968). Brief justification for our choice of montmorillonite additions is now included at line 437.

Regarding this reviewer's query about upscaling our results based on natural montmorillonite content and variation in marine surface sediments, to better emphasise the real-world impact of our findings, reviewer 2 (comment L21) similarly asks whether we could speculate about what our results might mean for methane production if montmorillonite concentrations increased beyond those used in our experiments, and where geographically this might be most relevant. In response to these reviewer queries, it is certainly clear from our results that methane production is increasingly inhibited with increasing montmorillonite concentrations (from 10 to 40 g/L), while as shown above, montmorillonite concentrations in marine surface sediments have been reported to reach near 130 g/L off the Coast of Peru in the South East Pacific Ocean (a site specifically studied for MAs biogeochemistry by Wang and Lee, 1990) and over 200 g/L in pelagic sediments of the South Pacific Ocean. As such we can confidently suggest that methanogenesis might be further inhibited with increasing montmorillonite concentrations, and thus that this process might be most important in geographical locations with high montmorillonite, like the South Pacific Ocean. Given that the turnover rate of cryptic methane cycling in marine surface sediment is comparable to the methane benthic flux out through the sediment-water interface (Xiao et al, 2017), the inhibition of this process potentially effects methane exchange between sediment and seawater and thus overall carbon cycle budgets. We are reluctant however, to assign an estimate to how much methanogenesis might be further inhibited with increasing montmorillonite concentrations, because marine surface sediments are more complex than our experiments here, and moreover, the methanogenesis inhibition with increasing montmorillonite concentrations that we measure is not strictly linear (e.g., if the relationship between montmorillonite concentrations and the reduction in final methane production was linear then the reduction in methane production at 40 g/L should be four times that at 10 g/L and this is not the case for DMA ($21.2 \pm 0.04\%$ reduction at 40 g/L $\neq 4 \times 4.0 \pm 0.2\%$ reduction at 10 g/L) or TMA ($24.9 \pm 0.3\%$ reduction at 40 g/L $\neq 4 \times 7.8 \pm 0.3\%$ reduction at 10 g/L) (see line 106). We have therefore expanded on the real-world impact of our findings at the end of the discussion at line 316; but we have emphasised that "Further research is required however, to investigate the relationship between montmorillonite concentrations, inhibition of methylotrophic methanogenesis and OC cycling, under more complex experimental conditions and *in situ* in marine surface sediments."

(In response to reviewer 1 comment 4, we also better emphasise the real-world impact of our findings, by highlighting that our "...slow-down of methane production in the presence of clay of up to ~2 days (Fig. 2) is comparable to the turnover rates of methane in marine surface sediments (Xiao et al, 2017) and our results therefore suggest that the adsorption of MAs to clay could substantially affect the cryptic methane cycle and thus local methane cycling in marine surface sediments." at line 224.)

3. Can you clarify in the text how the culture medium compares to what might be expected for a natural marine system (ionic strength, pH, etc.)? In particular, how might competition for adsorption sites differ in the real world in the presence of more organic compounds, especially within sediment porewaters? This seems relevant particularly when trying to assess whether the inter-layer migration that is proposed (i.e. Line 252) will be significant within a natural setting.

Response: Done. *Methanococcoides methylutens* TMA-10 used in this study was initially isolated from marine sediment and as such, the chemical composition of the culture medium (recipe from DSMZ Deutsche Sammlung von Mikroorganismen und Zellkulturen GmbH) mimics the chemical composition of seawater. The calculated ionic strength of the culture medium is thus ~0.5 M, approximating natural seawater (0.6 – 0.7 M), while pH is ~7.0, which is within the range of marine sediment porewater pH (6.9 – 8.3) (Ben-Yaakov, 1973; Reimers et al., 1996). The culture medium is optimised for cell growth under laboratory conditions (Sowers and Ferry, 1983, 1985) and is therefore nutrient-richer than porewater, and actually contains a higher concentration of organic substrates (mM range) than porewater (typically μ M range). We therefore expect that competition for adsorption sites in the real world, in the presence of typically lower concentrations of nutrients and organic compounds, is likely to be less than in our laboratory experiments, and is unlikely to inhibit interlayer migration of MAs in a natural setting. Comparison between the culture medium and natural marine sediment porewater is now included at line 360.

4. What are the implications in a natural system of a slowdown of methane production on a scale of a day or two? Do the authors believe this would impact the local methane cycling? Based on the plots, the change in timing is more significant than the change in final production due to the addition of clay.

Response: Done. The turnover rates of methane in marine surface sediment are actually very fast, especially in the top 0-2 cm of Aarhus Bay sediment for example (Xiao et al, 2017), which is one of the specific study sites at which low porewater but high sediment MAs concentrations have been observed (Zhuang et al., 2017). At this site the calculated turnover time of methane (methane concentration/methane oxidation rate) is 1 – 9 days, so a slowdown of a day or two in our laboratory experiments implies that the adsorption of MAs to montmorillonite in marine sediments could substantially affect local methane cycling in marine surface sediment. This information is now included at line 224. (Furthermore, under laboratory conditions, *M. methylutens* is necessarily grown under nutrient-rich conditions, which speeds up the growth rate compared to natural sediments. In a

nutrient-limited environment it is possible that the inhibition of final methane production observed in our experiments (~25% reduction in final methane production) will be enhanced to some extent.)

Reviewer #2 (Remarks to the Author):

Review of Xiao et al. “Mineralogical control on cryptic methane cycling in marine surface sediments” (NCOMMS-21-31580-T)

Synopsis

The primary focus of this study is to probe and quantify the mechanisms by which mineral protection may regulate organic matter (OM) preservation. In particular, the authors focus on methylotrophic methanogenesis and the role of mineral adsorption in regulating the concentration of methylamines (MAs), the dissolved substrate. Using a suite of batch incubation experiments (and abiotic controls), the authors show that MA adsorption to minerals both slows down methane production (by MA adsorption-desorption reactions) and decreases the final methane concentration (by the formation of a non-exchangeable, clay-bound MA fraction). The authors conclude that this provides the first mechanistic insight into how mineral interactions protect OM, and they speculate how this may relate to other mineral-OM processes and the global carbon cycle. Overall, I find this study to be highly topical and relevant to important carbon-cycle questions. (I note, however, that I am not a synchrotron expert and I therefore hope that other reviewers can better comment on this aspect of the manuscript.) It is well-written and articulated, and it poses a testable, mechanistic hypothesis for mineral-OM interactions. Still, I do have some “larger” comments and concerns, which I articulate below followed by line-item comments. Once these issues are fixed/clarified, I support publication of this manuscript in Nature Communications. Please do not hesitate to contact me for further details regarding this review.

Larger comment

1. Yeast extract compounds and the known promotion of methanogenesis by MAs: This question stems somewhat from my own ignorance, but how are the authors sure that there are no methylamines in the yeast extract added to the media? Similarly, how can the authors be sure that it is indeed the added MAs, rather than some other methyl compounds in the yeast extract, that fuels methanogenesis and is adsorbed to the clays? (I suppose the differences between panels in Fig. 2 support this claim). Have the authors performed (clay-free) control experiments whereby they can show that methane production scales in a dose-dependent manner with the amount of MMA, DMA, and TMA added, including no methanogenesis when no MAs are added (i.e., a zero intercept)? These experiments and results could be clearly articulated, giving more confidence to the experimental setup described here.

Response: Done. We can be sure that there were no MAs in the yeast extract (or in the trypticase peptone) added to the growth medium, and that there were no other methylated compounds in the yeast extract (or in the trypticase peptone) that could fuel methanogenesis because we did indeed perform clay-free control experiments in the absence and presence of MAs, to check for the inherent presence of MAs or other methylated compounds in the growth medium. These clay-free control experiments involved only the growth medium and the addition of *M. methylutens* in the absence and presence of TMA, during which we monitored methane accumulation and cell growth over 144 hours. As the graphs below show, we detected no methane accumulation or cell growth over the incubation period (near zero intercepts for both methane accumulation (~0.0001) and the growth curve (~0.01)). As *M. methylutens* is an obligatory methylotrophic marine methanogen, it can only use small, methylated molecules, like MAs, for methane production (Sowers and Ferry, 1983), thus the absence of methane accumulation and cell growth in the experiments without TMA demonstrate that there were no MAs or other methylated compounds in the yeast extract. (We also transferred 0.2 mL of the control experiment inoculum into 10 mL of fresh growth medium and observed no methane accumulation or cell growth over the incubation period). Information about these control experiments has been added at line 366 and the graphs below have been added to the Supplementary Information as Supplementary Fig. 4.

2. Methods details: In general, much of the Materials and methods section would benefit from additional clarification and details. I list here several examples:

Response: Done. We apologise for the lack of details in the Materials and methods section, and we now provide these below and in the main manuscript.

(i)

Were clays also sterilized for equilibration experiments (L322-331)? **Yes, they were autoclaved, now detailed at line 387.**

How are the authors sure that 24 hours is enough time for equilibrium to be reached? Could this be tested with shorter- or longer-duration equilibration tests? Initially we followed Wang and Lee (1993) (see Fig. 1 in their manuscript) who show that 2 hours is enough time for equilibrium in very similar experiments, but we also independently confirmed this, because we performed both clay-free and clay-added abiotic (inoculum-free) control experiments for our methanogen-clay interaction experiment. Our clay-added abiotic control experiments involved only the sterile growth medium, the addition of MAs at 30 mM and the addition of clay, and we monitored MAs concentrations in the medium at 24 hours (initial medium) and 144 hours (end medium) (and also in the exchangeable and non-exchangeable pools at 144 hours). These results are now included in Supplementary Fig. 1 (experiment Mnt10, Mnt20 and Mnt40 shown with pink, green and orange bars, respectively) and show that there was no significant change of MAs concentrations in the medium between 24 hours and 144 hours, so we are confident that 24 hours is long enough for equilibrium. Reference to Wang and Lee (1993) has been added at line 394. Details of the clay-added abiotic control experiments have been added at line 423.

How are the authors sure the total amount of dissolved MAs lost all becomes adsorbed to clays, rather than being adsorbed to the glass vial or being degraded? Did the authors perform a clay-free control? Could this be confirmed by quantifying the exchangeable and non-exchangeable adsorbed MAs and doing a mass balance? Yes we did perform clay-free control experiments as part of our abiotic control experiments for our methanogen-clay interaction experiments, as mentioned above. These allowed us to check for the loss of MAs via adsorption to the vials or degradation. Our clay-free control experiments involved only the sterile growth medium and the addition of MAs at 30 mM, and we monitored MAs concentrations in the medium at 24 hours (initial medium) and 144 hours (end medium). These results are now included in Supplementary Fig. 1 (experiment K, shown with black bars) and show that there was no loss of MAs from 30 mM added at time 0, to 30 mM in the initial medium at 24 hours to 30 mM at the end of the experiment after 144 hours. Details of the clay-free abiotic control experiments have been added at line 423.

As the reviewer suggests, no loss of MAs from the 30 mM added at time 0 can also be confirmed in the clay-added control experiments in Supplementary Fig. 1 (experiment Mnt10, Mnt20 and Mnt40 shown with pink, green and orange bars, respectively), because mass balance does indeed show that the sum of the concentration of MAs in the end medium at 144 hours, the exchangeable pool at 144 hours and the non-exchangeable pool at 144 hours equates to the 30 mM added at time 0. Details of the mass balance for MAs have been added at line 456.

(ii)

For both the ion- and gas-chromatography, what standards were used? How frequently? For the ion chromatography used to measure MAs, there are no certified standards for MAs, so we made standards using the same MA chemicals as we used for our experiments, and we injected these every 10 samples, now detailed at line 454. For the gas chromatography for methane, we used a gas mixture supplied by Air Liquide (France), and we injected this every 20 samples, now detailed at line 467.

What is the analytical precision? RSD for MAs detection via ion chromatography was < 3%, and for methane detection via gas chromatography was <2%, now detailed at line 456 and 468, respectively.

Which column (for gc) was used? A HayeSep Q 80/100 column, now detailed at line 467.

Were samples injected in triplicate, or was precision determined some other way? Yes, precision was checked by triplicate injection of standards, now detailed at line 485.

Similarly, please provide further details about the ferrozine assay and the ICP-MS methods. Dissolved Fe(II) in the medium was measured by Ferrozine assay and $(\text{NH}_4)_2\text{Fe}(\text{SO}_4)_2 \cdot 6\text{H}_2\text{O}$ solution was used as standard every 20 samples, with a RSD of <2%. Also dissolved Al^{3+} was measured using the Inductively Coupled Plasma Mass Spectrometry (Thermo Fisher Scientific, USA). Standards were injected every 10 samples, and Rhodium (m/z 103) at a concentration of 1 $\mu\text{g L}^{-1}$ was added to all standards and samples for use as an internal standard. The RSD was <2%. This information is now detailed at line 480.

3. Reporting of uncertainty: Throughout the manuscript text (in contrast to the figures), results are reported without any uncertainty. This uncertainty needs to be reported. This should include, for example, propagated analytical error when normalizing by surface area. Additionally, I disagree with the authors' assessment of using triplicate standard error (L399) throughout the study. While this is valid for assessing statistical differences between treatment means, lines such as (L14), "...reduces their concentration in the dissolved pool (up to $40.2 \pm \text{X.X} \%$)" and (L16), "reduces final methane produced (up to $24.9\% \pm \text{X.X}\%$)" should be reported with standard deviations since this is reporting uncertainty about a given number.

Response: Done. We thank the reviewer for bringing this to our attention, because our text at original line 399 stating that 'All data are expressed as the mean \pm standard error' was actually a typo – in

fact all data are expressed as mean \pm standard deviation (n=3). We apologise for this typo and have corrected the text at new line 521 and have also reported the standard deviations throughout the text as requested.

Line-item comments

L10 (repeated on L33, L181): The phrasing "...the link between mineral adsorption and retardation of microbial remineralization is almost entirely presumed..." is quite harsh to the existing literature. There are several publications (e.g., Keil et al. 1994 *Nature*, Vogel et al. 2014 *Nat Comms*, Lalonde et al. 2012 *Nature*, Keil and Mayer *Treatise on Geochemistry*, etc.) that do propose and test specific mechanisms of OC preservation by minerals. In contrast, it is probably fairer to say that the link is almost entirely presumed for the specific case of methanogenesis. Either way, the current study stands alone without phrasing such as this, and I recommend removing it.

Response: Done. Please accept our apologies – our wording was in no way intended to disregard the important work on mineral-OC interactions already reported in the literature; rather we were trying to highlight that whilst many of these existing studies propose and test specific mechanisms of OC preservation by minerals, very few actually conduct microbial incubations to determine whether these specific mechanisms are actually able to inhibit, or at least slow down, microbial remineralisation. This issue is recently highlighted in reference number 8 (Kleber et al., 2021, *Nature Reviews Earth & Environment*) which we cited at original lines 33 and 181. To avoid misunderstanding and an overly harsh reflection of the existing literature we have rephrased from "...the link between mineral adsorption and retardation of microbial remineralisation is almost entirely presumed..." to "...a direct link between mineral adsorption and retardation of microbial remineralisation is often presumed..." at lines 10, 31 and 188.

L21: "...and potentially other remineralization processes in marine sediments" is quite speculative and I suggest removing this—there are no data presented here that necessarily suggest this should be the case. (It may very well be true, just not related to the current study.) Still, methanogenesis is itself an important process and the paper again stands alone without this speculation. I instead suggest the authors end the abstract with more context on methanogenesis—e.g., what would it mean for methanogenesis if clay content increased or decreased? How would this influence overall carbon-cycle budgets? Where (geographically) is this process most important? The answers to some of these questions may also be somewhat speculative, but at least they would be grounded in the data presented here.

Response: Done. We have rephrased to remove speculation and focus on methanogenesis as suggested – from “Our data indicate that mineral-OC interactions strongly control cryptic methane cycling and potentially other OC remineralisation processes in marine sediments.” to “Our data indicate that mineral-OC interactions strongly control methylotrophic methanogenesis and potentially cryptic methane cycling in marine sediments.” With reference to the questions posed by the reviewer, our data show that methanogenesis is increasingly inhibited with increasing montmorillonite concentrations from 10 to 40 g/L, while montmorillonite concentrations in marine sediments have been reported to reach near 130 g/L off the Coast of Peru in the South East Pacific (a site specifically studied for MAs biogeochemistry by Wang and Lee, 1990) and over 200 g/L in surface pelagic sediments of the South Pacific Ocean (see response to reviewer 1, comment 2). As such we can confidently suggest that methanogenesis might be further inhibited with increasing montmorillonite concentrations, and thus that this process might be most important in geographical locations with high montmorillonite, like the South Pacific Ocean. Given that the turnover rate of cryptic methane cycling in marine surface sediment is comparable to the methane benthic flux out through the sediment-water interface (Xiao et al, 2017), the inhibition of this process potentially effects methane exchange between sediment and seawater and thus overall carbon cycle budgets. We thank the reviewer for encouraging us to speculate and have expanded on the real-world impact of our findings at the end of our discussion at line 316, but notably we have emphasised that “Further research is required however, to investigate the relationship between montmorillonite concentrations, inhibition of methylotrophic methanogenesis and OC cycling, under more complex experimental conditions and *in situ* in marine surface sediments.”

L48: should this read, “...small methylated OC compounds like monomethylamine (MMA), ...”

Response: Done. Corrected at new line 46.

L49: I don't really understand the phrase, “...which are not competitive substrates other than H₂ and acetate.” Please reword to clarify.

Response: Done. We have deleted this phrase as it is not relevant to the introduction.

L66-67 (and L292): The parenthetical, “(M. methylutens for abbreviation)” is unnecessary; it is common practice to simply use M. methylutens after writing the genus name upon the first instance.

Response: Done. Corrected at new line 64 and 347.

L76-92: How do these partition coefficients translate into moles of MA adsorbed per unit of mineral surface area? Looking at Table S1 and the authors' surface area estimates (L319), it looks like the surface area differences can explain some—but not all—of the difference in partition coefficients between minerals. Why is this the case? What other mineral-specific differences could explain this difference in adsorption affinity? I suggest reporting the area normalized partition coefficients here, but perhaps leaving some of this further discussion for the Discussion section.

Response: We thank the reviewer for their suggestion to calculate surface area normalised partition coefficients – there are now included in Supplementary Table 1. We leave reporting the raw partition coefficients in the results section because, as detailed above in our response to reviewer 1 comment 1, the surface area normalised partition coefficients help to make our case that there is another adsorption mechanism occurring for MSs adsorption to montmorillonite in addition to reversible adsorption of MAs – i.e., the relatively non-reversible adsorption that we measure (Fig. 3) and that we attribute to the occlusion of MAs in the montmorillonite inter layers - that causes the difference (elevation) in the montmorillonite partition coefficients compared to the other minerals. As the reviewer rightly suggests, this other adsorption mechanism is dependent on “other mineral-specific differences” because it requires the presence of an interlayer filled with easily exchangeable cation that are displaced by MAs. Of the clays we investigate only montmorillonite has such an interlayer. As such the elevated surface area normalised partition coefficients for montmorillonite compared to the other clays reflects both reversible adsorption and non-reversible occlusion into the interlayers. This extended discussion is now included at lines 261-282.

It is apparent that there also appears to be some differences in the surface area normalised partition coefficients between chlorite, illite and kaolinite, suggesting that these minerals may also have “other mineral-specific differences” that effect MAs adsorption. We feel that speculating on the reasons for the differences in the surface area normalised partition coefficients for the other clays however, is outside the scope of our manuscript, because, whilst we measure MAs adsorption to four clays, this is not our focus – rather we focus on MAs adsorption to montmorillonite and how this affects methanogenesis, and as such, we do not have NEXAFS for MAs adsorption to these clays, or data on the partitioning of MAs into an exchangeable vs. non-exchangeable pool, or indeed data on how the presence of these clays affects methanogenesis. Adsorption can be affected by a myriad of mineral specific differences other than surface area, including surface charge, presence of surface defects and roughness, presence of surface pore spaces, their size and tortuosity, the interaction of counterions in solution with the aforementioned properties, etc., and so we feel that measurement of at least some

of these properties would be required to more confidently comment on the chlorite, illite and kaolinite adsorption data.

L106: change “increase” to “increases”

Response: We think “increase” is correct? (new line 105) – we are referring to two phenomena – the slow-down of methane production and the reduction in final methane production, so we think “The slow-down of methane production and reduction in final methane production increase in the order MMA < DMA < TMA (Fig. 2).” is correct, but we are happy to change to “increases” at the Editor’s recommendation!

L125-128: Were there any exchangeable MAs detected in the abiotic control experiments? What about non-exchangeable (strongly adsorbed) MAs for the abiotic control experiments? Could the authors do a mass-balance for dissolved, exchangeable, and strongly adsorbed MAs for the abiotic control experiments to confirm that MAs are indeed partitioning as interpreted (see my larger comment, above).

Response: Done. Yes, there were exchangeable and non-exchangeable MAs detected in the abiotic control experiments, and the mass balance for these experiments is now shown in Supplementary Fig. 1, where it is clear that in the absence of *M. methylutens*, after 144 hours, the concentrations of MAs in the end medium plus the concentration of exchangeable MAs plus the concentration of non-exchangeable MAs equals the 30 mM MAs initially added to the experiment. Reference to Supplementary Fig. 1 is now included at line 123.

L209-212: I’m curious if the authors tried plotting the reduction in final methane concentration (i.e., the last time point in Fig. 2) as a function of the non-exchangeable MA concentrations (i.e., bars in Fig. 3)? If so, do they observe a linear relationship here? This should be the case, given the proposed mechanism. I think this would be a nice way to show that adsorption does indeed directly decrease methane production. (This is discussed somewhat beginning on L257. This discussion would benefit from the above-mentioned plot, because then the authors could point to a slope less than unity.)

Response: Done. We thank the reviewer for this comment - this is a great idea! We have made a new graph – Supplementary Fig. 3. We have normalised the data by percentage to produce a unitless slope, and then plotted the reduction in final methane production (%) (i.e., calculated from the last time point in Fig. 2) against the non-exchangeable MAs (%) (i.e., calculated from the bars in Fig. 3), and

we find that there is indeed a linear relationship ($R^2 = 0.93$). Reference to Supplementary Fig. 3 is now made at line 219. Reference to Supplementary Fig. 3 is now also made in the discussion at original line 257, now line 284, where we find that the slope is indeed more than unity, at 1.74...we assume the reviewer meant to say more than unity here, not less than unity – i.e., for every 1 unit of non-exchangeable MAs, there are 1.74 units of reduced final methane production – implying that there is an additional process limiting methane production, like Fe reduction.

Percentages of non-exchangeable MAs and reduction in final methane production were calculated based on mass balance (details of the mass balance calculations for MAs have been added at line 456 and for methanogenesis from MAs have been added at line 468):

*Percentages of non-exchangeable MAs = non-exchangeable MAs / (MAs in initial medium + MAs in end medium + MAs in exchangeable pool + non-exchangeable MAs) * 100% (ref. Fig 3)*

*Percentage reduction in final methane production = (methane production in clay-free treatment (K) - methane production in each treatment) / methane production in clay-free treatment (K) * 100% (ref. Fig. 2)*

L221 (and L226): Change Wang and Lee (1993) reference to Nature superscript format.

Response: Done.

L264-267: But this would be observable by a mass balance of MAs, correct? That is, if the release of ferrous iron inhibits methanogenesis, then the final concentration of MAs (whether in solution or adsorbed) should be higher than if no ferrous iron were released, right? In general, this Fe(II) discussion sounds somewhat speculative and could be improved with further mass balance details.

Response: Done. In our work we detected a release of Fe(II) into the medium, as plotted in Fig. 5, and an inhibition of methane production, as plotted in Fig. 2, but somewhat counterintuitively this was not expected to result in a higher final concentration of MAs than if no Fe(II) were released, because the inhibition of methanogenesis can occur through the diversion of electron flow to Fe(III) reduction (Bond, 2002; van Bodegom, 2004; Zhang 2013). As such, MAs are depleted during the bio-reduction of Fe(III) and subsequent release of Fe(II). To strengthen our discussion on the effect of Fe(II) release we have therefore highlighted that non-reversible adsorption of MAs into the interlayers of montmorillonite cannot account for all of the final reduction in methane production measured in our experiments because Supplementary Fig. 3 shows that “...for every unit of non-

exchangeable MAs there are >1 units of reduced final methane production.” (line 284). We have then clarified the following text to provide more background information on previous studies in which the bioreduction of Fe(III) and subsequent release of Fe(II) have occurred, and been suggested to inhibit methanogenesis.

L304: “...3.00 g trimethylamine...” I’m assuming this is only for the TMA experiments, correct? Was an equal amount of MMA or DMA added for the other experiments? This should be clarified.

Response: Done. Yes that is correct, for culturing *M. methylutens* the growth medium was made up with 3.00g (31.3 mM) TMA following the original recipe from DSMZ (new line 351), but for the methanogen-clay interaction experiments the growth medium was made up without TMA and then either MMA, DMA or TMA were added separately to reach 30 mM concentration. The preparation of the methanogen-clay interaction experiment has been clarified at line 401.

Figure captions: Please clearly state what is being shown for the error bars in each figure.

Response: Done. In all figures data points are presented as the mean and standard deviation of triplicate samples. All figure captions have been revised.

Figure 3: Does “initial medium” refer to the concentrations before inoculation but after the 24-hour equilibration? Similarly, the “Exchangeable” and “Non-exchangeable” results refer to those after the 6-day incubation, correct? Please clarify.

Response: Yes “initial medium” refers to the concentrations before inoculation but after the 24 hour equilibration; and “Exchangeable” and “Non-exchangeable” results refer to those after the 144 hour (6 day) incubation; apologies for the confusion, this is now clarified in the Fig. 3 caption.

Figure 4: Does this figure refer to the MAs adsorbed onto montmorillonite during the abiotic control experiment or after the incubations? Please provide additional details about the material being analyzed here. This is important because the authors then discuss a mineralogical change after incubation. How would these NEXAFS spectra look after incubation (assuming Fig. 4 currently shows results for abiotic adsorption experiments)?

Response: Done. Please accept our apologies for the confusion here. The reviewer is right in that the NEXAFS data shows results for the abiotic adsorption experiments – this is now clarified when we

describe the NEXAFS results at line 149, in the NEXAFS Fig. 4 caption at line 162, and in the methods at line 489. We also clarify in the discussion that we “...investigate the MA-clay system (Fig. 4) and the MA-clay-*M. methylutens* system (Fig. 5) at a mechanistic level.” at line 207. The reviewer is also right in that it could be important that the NEXAFS probe the adsorption mechanism of MAs to montmorillonite in the absence of *M. methylutens* but the XRD data show that the montmorillonite undergoes a mineralogical change in the presence of *M. methylutens* – so we need to be clearer about what this mineralogical change is, and whether it could affect the adsorption mechanism. First and foremost we omitted to report that no new mineral phases are formed after incubation, this is now added at line 173 – as such the mineral is still montmorillonite but the 001 peak is shifted to lower d-spacing and is wider and less intense. In our discussion we now also add that previous work similarly reports “...a concomitant shift to lower d-spacing together with a broadening and reduction in intensity of the 001 peak in the X-ray diffraction pattern (Zhang et al., 2013).” at line 294. We go on to say that “Previous work indicates that the peak changes reflect a partial collapse of the montmorillonite interlayers, in which the proportion of interlayers with $\sim 17 \text{ \AA}$ d(001) spacing decreases and those with $\sim 13 \text{ \AA}$ d(001) spacing increases (Zhang et al., 2013). A partial collapse of the interlayers is unlikely to substantially affect the reversible adsorption of MAs via electrostatic and hydrogen bonds or restrict the occlusion of molecules with small molecular size like MAs into the interlayers, but the presence of Fe(II) is suggested to contribute to slowed methane production and inhibition of final methane production...” at line 300. As such whilst we determine the adsorption mechanism of MAs to montmorillonite in the absence of *M. methylutens* and there is a mineralogical change in the montmorillonite in the presence of *M. methylutens*, this change is unlikely to substantially affect the ability of the mineral to adsorb MAs via the adsorption mechanisms that we identify.

(Unfortunately it is not possible to perform NEXAFS after incubation because the concentrations of MAs that remain adsorbed to the montmorillonite (i.e., only the relatively non-reversible occluded fraction – at best only 15% of the originally added MAs (line 135)) are too low to collect useful NEXAFS spectra. Whilst the presence of the microbes do have a minor effect on the structure of the montmorillonite, this is very unlikely to alter the adsorption mechanisms of the MAs, because the structure is not substantially changed, and also, we see that the distribution of the relatively non-reversible (non-exchangeable) MAs is essentially the same between the abiotic control in Supplementary Fig. 1 and the biotic experiment in Fig. 3).

Figure 5: Please include units for the y axis label.

Response: Done.

Reviewer #3 (Remarks to the Author):

Xiao and coauthors investigated the effect of mineral adsorption on the production of methane from methylamines (MAs), the process of which was reported earlier to drive the cryptic methane cycling in marine surface sediment. They found that MAs were strongly adsorbed into clay minerals. This adsorption significantly reduces methylamines concentrations in the dissolved pool and hence impacts methane production from a typical methylotrophic methanogen. They examined the adsorption of three MAs into four clay minerals and further tested the methane production by *Methanococcoides methylutens* TMA-10 with different concentrations of montmorillonite. This wonderful study provides the systematic evaluation of mineralogical control on microbial production of methane from MAs and improves our understanding in the interactions between physical, chemical and biological processes. However, the following comments need to be addressed before recommendation of publication.

General comments:

1. Title. The title is kind of misleading because the authors did not conduct experiments with any marine sediments. Although the methanogen *Methanococcoides methylutens* TMA-10 used in this study was initially isolated from marine sediment, there is no direct linkage between the results obtained from pure culture and cryptic methane cycling in marine surface sediments. I understand the authors intended to stress the potential implication, but this is not major finding that can be directly supported from the experiments. Also, methylotrophic methanogenesis can occur in both surface sediment and deeper sediments, and thus adsorption can affect methane production in deep sediment as well.

Response: Done. We take the reviewer's point here! and have revised the title as suggested to "Mineralogical control on methylotrophic methanogenesis and implications for cryptic methane cycling in marine surface sediment". We have kept the emphasis on marine surface sediment (as opposed to both surface and deeper sediment) because our experimental conditions are set up to mimic surface sediments (gdw clay / L, pH, ionic strength, etc) and so we feel that our results can be confidently applied to surface sediments.

2. The above comment also arises another question. I wonder why the authors did not use marine sediments to test the adsorption of MAs and methanogenesis. I think the slurry they made was far away different from real sediments, e.g., the porosity, MAs concentrations.

Response: Done. This comment relates to the comment made by reviewer 1 comment 2, in which we detail that montmorillonite in marine surface sediments has a reported global range between ~2 – 195 gdw montmorillonite/L in surface continental shelf sediments and ~65 – 215 gdw montmorillonite/L in surface pelagic sediments, and a range between ~8 – 129 gdw montmorillonite/L at the sites specifically studied for MAs biogeochemistry. As such we feel that our slurry with a range of 10 – 40 gdw montmorillonite/L experiments is reasonable representation of real sediments. Brief justification for our choice of montmorillonite additions is now included at lines 437. In response to reviewer 1 comment 3 we also detail that our experimental conditions of ionic strength and pH similarly represent marine surface sediments; now included at lines 223. So we are confident that our slurry was not too far away from real sediments. But we did not use marine sediment to test the adsorption of MAs and methanogenesis because our focus was to provide a new mechanistic understanding of MAs adsorption to clays and whether and to what extent this adsorption can control methanogenesis. Unfortunately real sediments are somewhat of a ‘black box’ and are difficult to characterise in terms of their mineralogy, and even more difficult to analyse at the macro and especially microscopic scales due to their inherent heterogeneity - for example, we would have been unable to perform NEXAFS on real sediments to the level of detail that we present in our experimental system, due to the complexity of the sediment matrix. As such using real sediment would have restricted us to macroscopic observations about the bulk uptake of MAs, and how changes in the bulk clay fraction (i.e., changes in the amount of the clay fraction rather than specific clay minerals) affects methanogenesis, and we would have struggled to find any mechanistic link between the two. So instead we chose to build on this bulk work that has already been done for MAs adsorption in real sediment (e.g., Wang and Lee, 1993) – indeed it is this previous bulk work that prompted us to investigate the relationship between MAs, clay and methanogenesis.

Other comments:

Line 83: The MAs concentrations showed in Fig.1 were not the same as 0.03-30 mM here (also as described in Line 325).

Response: Done. We apologise for the confusion here – the concentrations shown in Fig. 1 *are* the same as 0.03 – 30 mM stated at new line 80 and 392 (i.e. initial concentrations of MAs at 0.03, 0.3,

3, 10 and 30 mM), but the 0.03 and 0.3 mM data points overlap such that it appears there is only one data point near 0 mM initial MA, followed by the other data points at 3, 10 and 30 mM initial MA. We have added some clarification to the Fig. 1 caption.

Line 175: Unit is missing for dissolved Fe in Fig. 5.

Response: Done.

Line 192: I am confused here. What is the range of particulate matter concentrations? Does particulate matter mean particulate carbon? I think the slurry made with 10-40 g L⁻¹ clay were much more diluted than marine sediments. Why the concentrations were comparable?

Response: Done. This comment was also made by reviewer 1 comment 2 and relates to reviewer 3 comment 2 above – we apologise again for our references at original line 193, which refer to particulate matter concentrations and are therefore difficult to relate to a mass of montmorillonite per volume of water. As detailed above in our response to reviewer 1 comment 2 and reviewer 3 comment 2, with a global range between ~2 – 195 gdw montmorillonite/L in surface continental shelf sediments and ~65 – 215 gdw montmorillonite/L in surface pelagic sediments, and a range between ~8 – 129 gdw montmorillonite/L at the sites specifically studied for MAs biogeochemistry, we feel that our range of 10 – 40 gdw montmorillonite/L for our methane production experiments is reasonable. Brief justification for our choice of montmorillonite additions is now included at lines 437.

Line 270-273: This is speculative. I think no evidence existed for concurrent production and oxidation of methane by *Methanococcoides methylutens* TMA-10.

Response: Done. This sentence has been removed.

References cited in Response

Ben-Yaakov, Sam. pH buffering of porewater of recent anoxic marine sediment. *Limnology and Oceanography*, 1973, 18 (1): 86-94.

Bond DR, Lovley DR. Reduction of Fe(III) oxide by methanogens in the presence and absence of extracellular quinones. *Environ. Microbiol.* 4, 115-124 (2002).

van Bodegom PM, Scholten JCM, Stams AJM. Direct inhibition of methanogenesis by ferric iron. *FEMS Microbiol. Ecol.* 49, 261-268 (2004).

Chamley H. *Clay Sedimentology*. Springer, Berlin, Heidelberg (1989).

Griffin JJ, Windom H, Goldberg ED. The distribution of clay minerals in the World Ocean. *Deep Sea Research* 15, 433-459 (1968).

Jensen J.B. and Bennike O. Geological setting as background for methane distribution in Holocene mud deposits, Aarhus Bay, Denmark. *Continental Shelf Research*, 29, 775-784 (2009).

Martin, K. M., Wood, W. T., and Becker, J. J. A global prediction of seafloor sediment porosity using machine learning, *Geophys. Res. Lett.*, 42, 10640– 10646 (2015).

Reimers CE, Ruttenger KC, Canfield DE, Christiansen MB, Martin JB. Porewater pH and authigenic phases formed in the uppermost sediments of the Santa Barbara Basin. *Geochim Cosmochim Acta*, 1996, 60, 4037-4057.

Sowers KR, Ferry JG. Isolation and characterization of a methylotrophic marine methanogen, *Methanococcoides methyluten* gen. nov., sp. nov. *Appl. Environ. Microbiol.* 45, 684 (1983).

Sowers KR, Ferry JG. Trace metal and vitamin requirements of *Methanococcoides methylutens* grown with trimethylamine. *Arch. Microbiol.* 142, 148-151 (1985)

Tenzer R. and Gladkikh V. Assessment of density variations of marine sediments with ocean and sediment depths. *The Scientific World Journal*, 8, 823296 (2014).

Wang X-C, Lee C. Adsorption and desorption of aliphatic amines, amino acids and acetate by clay minerals and marine sediments. *Mar. Chem.* 44, 1-23 (1993).

Xiao K-Q, Beulig F, Kjeldsen KU, Jørgensen BB, Risgaard-Petersen N. Concurrent methane production and oxidation in surface sediment from Aarhus Bay, Denmark. *Front. Microbiol.* 8, (2017).

Zhang J, Dong H, Liu D, Agrawal A. Microbial reduction of Fe(III) in smectite minerals by thermophilic methanogen *Methanothermobacter thermautotrophicus*. *Geochim Cosmochim. Acta.* 106, 203-215 (2013).

REVIEWER COMMENTS

Reviewer #2 (Remarks to the Author):

The authors have thoroughly addressed my previous comments and concerns, and I therefore recommend publication of this manuscript in its current form.

Reviewer #3 (Remarks to the Author):

I am happy to see that my comments were well addressed in the revised manuscript. I recommend the publication of it.